The impact of climate change on flow conditions and wetland ecosystems in the Lower Biebrza River (Poland)

Mirosław-Świątek Dorota 1 D.Swiatek@levis.sggw.pl
http://orcid.org/0000-0002-6000-9758 Marcinkowski Paweł 1
Kochanek Krzysztof 2
http://orcid.org/0000-0002-9735-2103 Wassen Martin J. 3
1 Department of Hydrology, Meteorology and Water Management, Institute of Environmental Engineering, Warsaw University of Life Sciences—SGGW , Warsaw , Poland
2 Institute of Geophysics, Polish Academy of Sciences , Warsaw , Poland
3 Copernicus Institute of Sustainable Development, Utrecht University , Utrecht , Netherlands
Fu Guobin
Electronic publication date: 2020 Sep 11
Publication date: 2020
Volume: 8
Electronic Location ID: e9778
Received 2020 Mar 3; Accepted 2020 Jul 30
Copyright: © 2020 Mirosław-Świątek et al.
Copyright year: 2020
Copyright holder: Mirosław-Świątek et al.
License: This is an open access article distributed under the terms of the Creative Commons Attribution License, which permits unrestricted use, distribution, reproduction and adaptation in any medium and for any purpose provided that it is properly attributed. For attribution, the original author(s), title, publication source (PeerJ) and either DOI or URL of the article must be cited.
License URL: https://creativecommons.org/licenses/by/4.0/

Keywords: SWAT, Climate change, Trend analysis, The Biebrza river

Funding: Polish National Agency for Academic Exchange PPI/PZA/2019/1/00107/U/00001 This article was supported by the Polish National Agency for Academic Exchange under Grant No. PPI/PZA/2019/1/00107/U/00001.

==============================
Water plays a key role in the functioning of wetlands and a shortage or contamination of it leads to changes in habitat conditions and degradation of ecosystems. This article scrutinizes the impact of climate change on the hydrological characteristics of floods (maximum flow, duration, volume) in the River Biebrza wetlands (North-East Poland). We analysed the trends in duration and volume of flood and maximum discharges in the historical period 1970–2000 and predicted these for the future periods 2020–2050 and 2070–2100, respectively. Next we assessed the impact on the wetland ecosystems. The basis of our assessments consists of statistical analyses of hydrographs and calculations by the Soil and Water Assessment Tool hydrological model and considering nine bias-corrected climate models. The results indicate that both volume and duration of winter floods will keep increasing continuously under Representative Concentration Pathways 4.5 and 8.5. The reduction in peak annual floods is expected to decline slightly in both scenarios. On the other hand, the analysis of trends in mean and standard deviation revealed negligible tendencies in the datasets for summer and winter hydrological seasons within the three time frames analysed (1970–2000; 2020–2050; 2070–2100). We foresee several future implications for the floodplain ecosystems. Shifts in transversal ecosystem zonation parallel to the river will likely take place with more highly productive flood tolerant vegetation types. Nutrient availability and algal blooms during spring inundations will likely increase. Slowdown of organic matter turnover later in summer will lead to a higher peat accumulation rate. Logistical problems with summer mowing and removal of bushes in winter may enhance shrub encroachment.

Introduction

Wetlands, due to their functions in biodiversity and landscape preservation, formation of habitats for many endemic plant and animal species, and also in reducing the outflow of groundwater into rivers thus providing protection from erosion and floods, retaining groundwater and surface water, water purification and accumulation of organic matter, are among the most valuable natural systems on Earth and belong to the most complex ecosystems (Costanza et al., 1997, 2014). Water plays a key role in the functioning of wetlands, and a shortage or contamination of it leads to changes in habitat conditions and the degradation of ecosystems. An exploitative approach to the management of natural resources, which concentrated on using these ecosystems for production purposes only (downgraded by drainage in order to convert it into agricultural land or by wasteful exploitation of peat), resulted in many wetlands becoming endangered and losing their original functions (Joosten & Clarke, 2002). The roots of the wetlands’ natural dysfunction may stem not only from the evident anthropo-pressure, but also from climate change, which is a serious threat potentially leading to transformation of hydrological conditions (Dorau et al., 2015).

The activities aimed at restoring the selected functions of transformed wetland ecosystems or maintaining a good condition and ecological balance of properly functioning ecosystems are nowadays the issues at stake related to the environmental management process. Therefore, it is extremely important to consider the interplay of pressures resulting from human activities, climate change and ecological dependencies (Mitsch, 2014). Quantifying these elements, forecasting their variability in the future, as well as taking into account their impact on ecosystems still poses a problem. Although the impact of climate change on wetland hydrology has been studied this is mostly done on a global, continental or national scale (Hoegh-Guldberg & Bruno, 2010; Witte et al., 2012; Finlayson, 2013; Thompson et al., 2009) and not targeted at specific wetland areas of interest. Therefore it remains challenging to assess the impact of climate change on wetland hydrological characteristics and on the occurrence and functioning of ecosystems in wetlands of special interest.

The River Biebrza Valley, where the river has retained its natural lowland character, is part of the largest peat complex in Central Europe. It is one of the most valuable and demanding (in terms of marsh management) ecosystems in Europe. The Biebrza fens and marshes function as a reference area for activities aimed at restoration of wetlands for many transformed river valleys in Europe (Bootsma, Wassen & Jansen, 2000; Wassen et al., 2006). The regular occurrence of spring floods in the lower Biebrza basin (LBB) is the main factor conditioning the development and functioning of diverse wetland ecosystems. The valley inundation occurs almost every year and lasts from several weeks to several months. The origins of these inundations are twofold: spring thaws and melting of snow and high precipitation (Kiciński & Byczkowski, 1983). Berezowski et al. (2018) analysed long-term (1960–2012) changes in meteorology, hydrology, soil and vegetation, and also conservation history and found significant trends indicating climate change: increases in temperature and evapotranspiration and earlier start of spring floods. They also identified (1970–2000) a shift towards drier vegetation types, but also found that local restoration measures such as blocking drainage via ditches, not clearing aquatic vegetation, bush removal and mowing that were implemented have mitigated climate change effects and have led to increases in soil moisture and wetter vegetation types in some areas.

The River Biebrza’s abundant spring floods determine the development of wetland plants and shape rich feeding grounds, especially for wetland birds which make the Biebrza Valley the most valuable bird refuge in Europe as a nesting and feeding area and during migration (Maciorowski et al., 2014). The occurrence and zonation of wetland ecosystems with a fluviogenic type of supply is associated with the dynamics of floods, which provide a diversity of habitats depending on the duration and extent of the flood events (Chormański, Mirosław-Świątek & Michałowski, 2009).

To quantify the impacts of climate change on regional hydrological regimes, the outcomes of climate models coupled with hydrologic models are commonly used (Arnell & Gosling, 2013). One of the most popular hydrological models used in the assessment of climate change impact on different sectors, including environment, is the Soil and Water Assessment Tool (SWAT) model. It has been applied in numerous studies investigating climate change impact on wetlands ecosystems hydrology and well-being (Zhang et al., 2011; Kundzewicz et al., 2018; O’Keeffe et al., 2019).

The issues related to the quantitative assessment of climatic pressures and their role in the management of ecosystems in the Biebrza Valley have not been widely discussed in the literature. So far, the main focus has been on the pressure related to agricultural use of these areas (Swiatek et al., 2008). Grygoruk et al. (2014) analysed the changes in average monthly precipitation in this area in a simplified way based on historical observations and 10 separate forecasts of climate changes in the Biebrza Valley for 2070–2100 with different combinations of General Circulation Models (GCM) and Regional Climate Models (RCM). They describe the potential effects of the most negative scenario of climate change—the possibility of winter drought (no snow accumulation, low rainfall) and hot summers with high episodic rainfall separated by weeks of droughts. The use of these forecasted climate changes by the Biebrza National Park to implement an adaptive management model was proposed in Grygoruk & Rannow (2017).

The multiannual tendencies of climatic changes (e.g. mean annual precipitation, temperature and snow cover) and variability of hydrologic characteristics (annual maxima in winter seasons and water discharge and stage in summer seasons) in the Biebrza River basin in 1966–2003 were analysed using the Mann–Kendall test by Maksymiuk et al. (2008).

In riparian wetlands, the extent and duration of floods keep floodplain ecosystems in appropriate hydrological and geochemical conditions (Chormański, Mirosław-Świątek & Michałowski, 2009; Chormański et al., 2011; Swiatek et al., 2008). We will show that these hydrological indicators in the LBB are closely related to the characteristics of floods observed at the Osowiec water gauge. The aim of the current paper is to assess the impact of climate change on (1) the hydrological characteristics of floods (duration, volume, maximum flow) and on (2) specific wetland ecosystems. For this we carried out statistical analyses of the hydrographs derived from the SWAT hydrological model for the historical period 1970–2000 and future time horizons 2020–2050 (near future (NF)) and 2070–2100 (far future (FF)), considering nine bias-corrected climate models at the Osowiec water gauge. Next, we qualitatively assessed expected changes in ecosystems resulting from the estimated hydrological and subsequent other environmental changes.

Materials and Methods

Study area

The study was carried out in the northern part of the Lower Basin of the River Biebrza Valley, considered a well preserved, natural floodplain ecosystem representing main ecological features of such systems in temperate European lowlands (Fig. 1). This area (6,300 ha) is located in North-Eastern Poland and is characterized by a broad land depression filled with sand, on which peat soils were developed in the Holocene. The valley was shaped in the late Pleistocene by fluvio-glacial waters of the Vistulian Glaciation. The floodplain is covered by a mosaic of marshy vegetation ranging from sedge, sedgemoss and reed communities to willow shrubs, black alder forest, swampy birch and peat coniferous forest, and this area is among the most valuable wetlands globally (Wassen et al., 2006). The valuable ecosystems here are also represented by the large areas of open semi-meadows with near-natural vegetation, which are the result of agriculturally managed as extensively used meadows of a seasonal mowing regime. Among the dominant plant communities the four following alliances are of special interest in the floodplain: Caricetum appropinquatae (CA), Caricetum gracilis (CG), Glycerietum maximae (GM) and Phragmitetum communis (PC) (Fig. 1). These alliances are typical for temperate European floodplains that developed in peat-filled valleys and which represent unique biodiversity (Budka et al., 2019). The elevations in the study area vary from 103 up to 118.5 m a.s.l. (Mirosław-Świątek et al., 2016). The average annual air temperature is 6.6 °C (Banaszuk, 2004) and the average annual sum of precipitation is as high as 560 mm (Kossowska-Cezak, 1984). The Lower Basin is part of the most valuable bird habitat as a site of nesting, foraging and migration (Polakowski, Broniszewska & Goławski, 2014) and also as an area protecting valuable and rare species of birds residing in the wetlands (Maciorowski et al., 2014). The healthy floodplain ecosystem functioning in the Lower Biebrza is connected with the river Biebrza flooding regularly every year for at least several weeks or months (Kiciński & Byczkowski, 1983). Grygoruk et al. (2013), basing upon the results of national hydrological survey, noticed that when the discharge exceeds 25 m3/s at the Osowiec gauge, we can expect the floodplain to be flooded. In such a case the water depth in the floodplains may reach 1.5 m (Mirosław-Świątek et al., 2017). Wetlands of LBB depend on the interactions between seasonal flooding, groundwater discharge and rain (snowmelt) water accumulation (Berezowski et al., 2019; Chormański et al., 2011). But in the northern part of the basin, where our study area is located, the main mechanism of flooding is the river flood wave. Its hydrological characteristics are observed at the Osowiec gauge (Fig. 1) (Okruszko, 2005).

Figure 1 Case study (A - location, B - vegetation type).

SWAT model

The SWAT model is a process-based, continuous-time model which simulates the movement of water, sediment and nutrients on a catchment scale (Arnold et al., 1998). It is suitable for investigating the interaction between climate, land use and water quantity. The SWAT model enables simulation of long-term impacts of climate changes on water, sediment, and nutrient loads in catchments with varied topography, land use, soils and management conditions. The catchment area is divided into sub-basins and further into Hydrologic Response Units (HRU), which are the smallest spatially distinguishable units. Water balance, sediment and nutrient loads are calculated for each HRU. Runoff is aggregated at the sub-basin level and routed through the stream network to the main channel to indicate the total runoff for the river basin. In this study, we use simulations of the existing, extensively calibrated and validated SWAT model of the Vistula and Odra river basins (VOB), (Piniewski et al., 2017b). Piniewski et al. (2017b) conducted a multisite calibration (1991–2000) in 80 non-nested catchments (461–2,620 km2) in Poland used as ‘benchmark’ catchments. In the second step they clustered benchmark catchments using a large set of hydrological metrics in order to derive homogeneous calibration areas, from which calibrated parameters could be transferred to non-calibrated catchments regarding their hydrological similarity. Similar approach with transferring the parameters between sub-catchments regarding their geo-environmental similarities was performed by Choubin et al. (2019). VOB model has been validated in 30 catchments for period 2001–2010. The SUFI-2 algorithm (Abbaspour, Johnson & van Genuchten, 2004) within the SWAT-CUP version 5.1.6.2 software package was used for SWAT model calibration, sensitivity and uncertainty analysis. They used Kling-Gupta efficiency (KGE) as an objective function. In this study we used the simulated daily streamflow data (in m3/s) for the sub-catchment (extracted from VOB model) within which the Osowiec gauging station is located. Due to the fact that this particular sub-catchment was not subject to calibration and validation in VOB model, we calculated the KGE and R2 statistics to validate the SWAT model simulation accuracy vs observations in this sub-catchment. SWAT model performance as well as goodness-of-fit measures are presented in Fig. 2. It is worth noting that the SWAT model performs well in terms of streamflow simulation for both high and low flows for wet and dry years. The SWAT model captures the characteristic behavior of the catchment in which high spring flows are mainly determined by snowmelt processes. Figure 2B clearly shows that a thicker snow cover in winter correlates with a higher flow in spring. Also, a high annual precipitation causes an increase in low flows (Fig. 2A) which is especially evident in 2010 (one of the wettest years in Poland over the last few decades).

Figure 2 SWAT model simulations vs observed time series in Osowiec gauging station.

(A) Annual precipitation sum in the Biebrza catchment, (B) snow cover thickness in the Biebrza catchment.

Climate change scenarios

In this study, the SWAT was driven by climate forcing data from the Climate Change Impact Assessment for Selected Sectors in Poland (CHASE-PL project) Climate Projections: 5-km Gridded Daily Precipitation and Temperature Dataset (Mezghani, Dobler & Haugen, 2016), consisting of nine bias-corrected General Circulation Model–Regional Climate Model (GCM–RCM) runs (involving four different GCMs and four different RCMs) provided within the Coordinated Downscaling Experiment-European Domain (EURO-CORDEX) experiment projected to the year 2100 under Representative Concentration Pathway (RCP) 4.5 and RCP 8.5 (Piniewski et al., 2017a). All bias-corrected values (quantile mapping method by Norwegian Meteorological Institute, (Gudmundsson et al., 2012)) of precipitation and air temperature were available for three time periods: 1971–2000 (historical period), 2021–2050 (near future—NF) and 2071–2100 (far future—FF).

The projected mean annual temperature in the Biebrza catchment is expected to rise by approximately 1.2 °C in the NF and 2 °C in the FF under the RCP4.5, with notable seasonal variation: higher change in winter (2.5 °C in FF) and lower in summer (1.7 °C in FF). For the RCP8.5, the temperature increase reaches a mean of 3.6 °C in FF, whereas in NF, it is similar to RCP4.5 (1.3 °C versus 1.1 °C). The River Biebrza catchment is expected to receive more precipitation in the future in all seasons (with a much higher increase in winter and spring). In the RCP4.5 scenario, the projected annual mean precipitation increase is approximately 6% in NF and 10% in FF, while for RCP8.5, the projections show a 16% increase in FF (Piniewski et al., 2017a).

We analyzed the mean multi-annual values of investigated flood characteristics for each combination of RCP-time horizon-climate model and presented the results in boxplots. We also carried out a mean annual trend analysis. The statistical detection of the trends was carried out on three types of datasets: (i) annual peak flows above 25 m3/s (in m3/s), (ii) duration of the highest annual flood waves above 25 m3/s (in days), (iii) volume of the highest annual floodwaves above 25 m3/s (in m3); these datasets (i–iii) were considered for two RCPs: (a) RCP4.5 and (b) RCP8 and for three time horizons: (1) 1971–2000 (historical period), (2) 2021–2050 (near future—NF) and (3) 2071–2100 (far future—FF). SWAT simulated time-series in the Osowiec gauging station, including flows above 25 m3/s are available in the raw data (Discharge-model file) separately for each season (winter, summer), time horizon (NF, FF) and RCP (4.5, 8.5).

Trend analysis by means of statistical models—the method

To address the variability or regular trends of the river regime in the near future we employed the methodology developed for normally distributed datasets by Strupczewski & Kaczmarek (1998, 2001) and later extended to non-normally distributed datasets by Strupczewski et al. (2016). This methodology allows for estimation of the quantitative trends simultaneously in mean and standard deviation in SWAT simulated hydrological datasets (SWATmodel_QtimeSeries file in the Supplemental Materials). A thorough description of this methodology would fall beyond the scope of this paper, and can be found in Strupczewski & Feluch (1998), Strupczewski & Kaczmarek (1998, 2001); Strupczewski, Singh & Feluch (2001) and Strupczewski et al. (2016). However, the basics of this methodology deserve a few comments. The method involves the assumption that the annual river peak flows constitute mutually independent random variables and are the only source of information on the variability of hydrological processes. This means that the proposed approach does not consider the variability of climatic pressures, anthropogenic changes in the catchment and diversity in the regime resulting from water management policy in the modelled catchment. To be clear, we are perfectly aware that all these factors do influence the water flow in the rivers, but the statistical modelling method we used in our study concentrates on the results of these impacts forwarded to the river’s response and expressed in the annual peak flows datasets. We consciously resigned from non-parametric trend-detecting tests (e.g. Mann–Kendall Test For Monotonic Trend, Mann, 1945; Kendall, 1975; Gilbert, 1987) because they perform well when the changes are significant and the datasets have no long gaps in the time series (Hirsch, Slack & Smith, 1982). The gap-years in the data series occur when no discharge above 25 m3/s is recorded (i.e. no flood occurred) for that particular year; this is especially visible for the winter season, where gaps may last several subsequent years.

The proposed method consists of estimating the time-dependent parameters of selected statistical models by means of the maximum likelihood (ML) estimation method. The best statistical model was selected from seven competing statistical distribution functions:Two-parameter distribution functions: Normal (N), Gumbel (Gu),

Three parameter distribution functions: Log-Normal (LN), Pearson type 3 (Pe), Generalised Extreme Value (GEV), Generalised Log-Logistic (GLL) and Weibull (We).

These distributions were selected because of their confirmed applicability to model annual and seasonal peak flows in Polish rivers (Strupczewski et al., 2009, 2012; Kochanek, Strupczewski & Bogdanowicz, 2012). A detailed description of these distribution functions and their properties would be beyond the scope of this paper, however the algebraic forms of these functions and formulas of parameter estimation by of maximum likelihood method can easily be found in any statistical or hydrological handbook, for example in Rao & Hamed (2000). The criterion for selecting the best non-stationary statistical model for a given data series can be made by comparing Akaike Information Criterion (AIC) values (Akaike, 1974; Hurvich & Tsai, 1989).

The majority of the statistical models used in hydrology are characterised by the parameter set, namely location, scale and shape (three-parameter distribution functions). The two-parameter distributions used in this study (Normal and Gumbel) have location and scale parameters only. Although useful in terms of estimation and statistical analysis, these parameters hardly ever have a physical interpretation. What is more, the scale parameter, for example of one distribution function usually means something else than in another distribution function. In order to enable easy comparison between the results provided by each of the competing statistical models, their original sets of parameters (two or three, depending on the statistical model) of each distribution function were replaced by the statistical moments which are the usually non-linear functions of the regular parameters. This re-parametrisation of the distribution functions provides an opportunity to introduce the trends explicitly into the statistical moments, which complies with the common and intuitive methods of trend analysis.

We assumed that time variability will be introduced into the two first statistical moments, that is mean (μt) and standard deviation (σt), and that the third parameter, usually the shape parameter (if it exists), will be time independent. In consequence, the likelihood equations contain both time (t) dependent mean (μ(t) = mt) and standard deviation (μ2(t)1/2 = σt), so they would be solved jointly. In fact, one can assume any kind of mt = f(t) and σt = g(t) functions, depending on the time series, however the most common is the assumption of the linear dependence of the process on time: mt=at+b

σt=ct+d

where a, b, c and d—coefficients.

Note that in such notation a in mean and c in standard deviation are the trend coefficients. Therefore, to estimate the trends in mean and standard deviation by means of the maximum likelihood method we have to estimate five (for three-parameter distribution functions) or four parameters (for two-parameter distribution functions), that is a, b, c, d and k which is the shape parameter (if this exists). From a practical point of view, estimation of the parameters by the maximum likelihood method consists of solving a system of five non-linear equations (Eq. (1)) which can only be solved by numerical methods.

(1) {∂ln⁡L∂a=0∂ln⁡L∂b=0∂ln⁡L∂c=0∂ln⁡L∂d=0∂ln⁡L∂k=0

where:

ln⁡L=∑i=1Nln⁡f(xi,a,b,c,d,k)—logarithm of the likelihood function

f(xi,a,b,c,d,k)—probability density function (statistical model)

a, b, c, d, k—parameters of the statistical model (to be estimated)

xi—elements of the sample

N—number of elements in the sample

We developed software for estimating the linear trends in mean and standard deviation in seven distribution functions (Strupczewski et al., 2016). Apart from the values of the parameters that provide the maximum rate of the likelihood function, the software output also provides the values of the AIC that enable us to choose the statistical model that best suits the dataset used for calculation. The software is able to cope with the measurement gaps, provided that the spaces in datasets do not exceed a few missing values.

The impact of hydrological changes on the wetland ecosystems

In the LBB extents and durations of floods are considered key drivers of riparian ecological processes. Because our hydrological analyses are being developed for the Osowiec water gauge it is therefore necessary to find a link between the analysed flood characteristic at this gauge and floodplain inundation features. The relationships between annual maximum discharge and spatial range of the inundation waters as well as the duration of inundations and floodwave duration observed in the Osowiec gauge were developed based on the results of the 1D hydrodynamic flow model. This model and its results coupled with GIS analysis (inundation maps) were descripted in details in Chormański et al. (Chormański, Mirosław-Świątek & Michałowski, 2009). The maximum flood extent (defined as the number of flooded pixels in digital elevation model—DEM) for the daily time series from 1961 to 1996 and the duration of inundation in different plant communities (Fig. 1) in this period were used to analyse the relationship between these characteristics and variables observed for the Osowiec water gauge. Due to the importance of typical floodplain ecosystems and its biodiversity we in our analysis concentrate on CA, CG, GM and PC (Fig 1). A qualitative description of expected changes in ecosystems resulting from the estimated hydrological changes was carried out by comparing the estimated future hydrological and other environmental characteristics with typical conditions as observed in the distinguished vegetation communities and reported in literature.

Results

Inter-comparisons of SWAT model-based simulations of selected flood characteristics (i.e. duration, volume and maximum discharge) concerning the RCP and time horizon are presented separately for winter and summer seasons (Figs. 3 and 4). Such a distinction is dictated by the diverse genesis of floods, with the dominant role in their formation being played by melting snow in spring and rainfall in summer. For flood duration, a significant increase is projected, with a significant rise in the ensemble median from 80 days in the historical period to 105 and 135 in NF and FF, respectively, for RCP4.5. The increase escalates in RCP8.5 to 135 and 170 for NF and FF, respectively. A similar pattern is observed for flood volume, which increases with the time horizon and RCP. The ensemble median increases from 8 billion m3 in the historical period to 12 billion m3 in NF and 21 billion m3 in FF for RCP4.5, and for RCP8.5 to 21 and 42 billion m3 for NF and FF, respectively. However, this increase is mainly a result of the flood duration rather than the maximum discharges, which even tend to slightly decrease in the future (Fig. 3). The duration and volume of summer floods seem to follow the same pattern as for the winter season, although the magnitude of increase is much lower. For flood duration, the ensemble median increases from eight days in the historical period to 20 and 29 in NF and FF, respectively, for RCP4.5. For RCP8.5, the flood duration increases to 21 and 60 days for NF and FF, respectively. There is a slight difference in maximum discharge noted for the summer season compared to winter floods, as high flows increase with the time horizon and RCP. Comparing the values, the ensemble median of maximum flow increases from 25 m3/s to 39 m3/s and 41 m3/s for NF and FF, respectively. Comparing the RCPs, in this case the differences are negligible.

Figure 3 Projected changes in maximum discharge (A), flood duration (B) and volume (C) during winter season based on SWAT model simulations.

Light grey colour stands for actual greenhouse gas concentration, medium grey for RCP4.5 and dark grey for RCP8.5. Boxes denote—first–third quartile values, whiskers—non outlier range, circles—outliers, asterisk—extremes, rectangles—mean values and squares—median values.

Figure 4 Projected changes in maximum discharge (A), flood duration (B) and volume (C) during summer season based on SWAT model simulations.

Light grey colour stands for actual greenhouse gas concentration, medium grey for RCP4.5 and dark grey for RCP8.5. Boxes denote—first–third quartile values, whiskers—non outlier range, circles—outliers, asterisk—extremes, rectangles—mean values and squares—median values.

The preliminary calculations described in the previous chapter revealed that a certain dataset, that is annual maximal discharges, volume of the floodwaves and residence time tend to be represented by particular dominating statistical models. For instance, the datasets based on the maximum discharges are often appointed by the AIC to the group of the extreme value distribution functions (statistical models). In consequence, most often these are the Gumbel (Gu) or Log-Normal (LN) distribution functions which were predominately selected as the best for modelling such datasets, but in few cases the Normal (N) distribution function proved to be the best according to the AIC. Even though the criterion of the AIC occasionally pointed to another statistical model (e.g. GEV or We), the differences between the AIC values for the best statistical model (i.e. GEV or We) and the Gumbel one (usually the second in the competition) were negligible. Therefore, for brevity we decided to apply the Gumbel or Log-Normal statistical models (depending which of the two prevailed) to the datasets of annual peak flows. Similarly, the volume datasets are best described by the Gumbel or Log-Normal distribution functions. For the data of flood wave duration (in days) three statistical models, that is Normal, Gumbel or Log-Normal distribution functions, are most often indicated as suitable for the samples. Consequently the list of seven potential statistical models to choose from was reduced to the two two-parameter and one three-parameter distribution functions whose parameters are relatively easy to estimate (in terms of numerical calculations). On top of that, there is no need to estimate an additional shape parameter for two-parameter statistical models. This attempt to reduce the number of parameters to be estimated is a consequence of the general principle of avoiding the ‘overparameterisation’ of hydrological models and results in a reduced risk of numerical errors when optimizing the likelihood function (the larger the number of parameters, the higher the risk of algorithm failure). Note that all the calculations were carried out separately for summer and winter seasons, so the results were summarised in two thematic pairs of diagrams: Figs 5 and 6 for the summer, Figs. 7 and 8 for winter floods.

Figure 5 The distribution of the mid-term (14th year of the time series) mean values of: maximum discharge (A), flood duration (B), flood volume (C) and standard deviation of: maximum discharge (D), flood duration (E), flood volume (F), for RCP4.5 and RCP8.5 scena.

The solid line in the middle of the box is the mean value of the results for nine models. The dominating models for each calculation series are in brackets.

Figure 6 The distribution of the relative values of trends in mean: maximum discharge (A), flood duration (B), flood volume (C) and standard deviation of: maximum discharge (D), flood duration (E), flood volume (F), for the summer season.

The solid line in the middle of the box is the mean value of the results for nine models. The dominating models for each calculation series are in brackets.

Figure 7 The distribution of the mid-term (14th year of the time series) mean values of: maximum discharge (A), flood duration (B), flood volume (C) and standard deviation of: maximum discharge (D), flood duration (E), flood volume (F), for RCP4.5 and RCP8.5 scena.

The solid line in the middle of the box is the mean value of the results for nine models. The dominating models for each calculation series are in brackets.

Figure 8 The distribution of the relative values of trends in mean: maximum discharge (A), flood duration (B), flood volume (C) and standard deviation of: maximum discharge (D), flood duration (E), flood volume (F), for the winter season.

The solid line in the middle of the box is the mean value of the results for nine models. The dominating models for each calculation series are in brackets.

First we calculated the mean and standard deviation values for the mid-term year (i.e. the 14th year of the series) of the analysed 27-year data series (with gaps) in the baseline and subsequent RCP4.5 and RCP8.5 scenarios. The mid-term values of the mean (m14 = a14 + b) and standard deviation (σ14 = c14 + d) for the maximum discharge, duration and volume of the floodwave provide the base and summarising information about the possible changes of the particular properties of the floodwaves measured and modelled within the time. Of course, the estimation parameters allow the mean and standard deviation to be calculated for any time t within the dataset time frame, and also extrapolate them outside the time frames when necessary. However, extrapolation is not recommended far beyond the periods considered in the time series used for estimation, because this may lead to erroneous conclusions. Figures 5 and 7 illustrate the progress in the expected values and variability of maximum discharge, duration and volume for summer and winter seasons, respectively. The diagrams represent the median (thick solid line) and the distribution of the SWAT modelling results—the lower (first) and upper (third) quartiles (dashes) of the mid-term values calculated for nine climate models. The first 3 years were deducted from each dataset due to this being the warm-up time, so that the climatic models would adjust and produce credible results.

The values of the trends themselves provide information about the stability of a particular floodwave parameter within a 27-year data series. The results of the trend calculations are presented by box-whisker diagrams in Figs. 6 and 8. The box-whisker plots in the concise but informative graphic form show the basic summary statistics of trends in all nine climate models, that is median—thick solid line in a box, the distribution of the SWAT modelling results for nine climate models—the lower (first) and upper (third) quartiles are visualised by the bottom and top edges of the boxes, respectively, minimum and the maximum values—lower and upper whiskers, respectively, and outliers visualised by the circles. Each plot presents the situation for a different climatic scenario (RCP4.5 and RCP8.5) and for a different period in time (baseline, NF and FF). The graphs depict the relative trends in mean value and standard deviation (in %) for all three datasets. The trends (a or c) in time dependent mean (mt = at + b) and standard deviation (σt = ct + d) refer to the intercept parameters b and d, respectively, denoting the mean value and the standard deviation for t = 0. Such a relative presentation of the trends offers a possibility to analyse the effective trends in mean and standard deviation.

Case A. Summer season—Figs. 5 and 6

The summary statistics (visualised by the diagrams) confirm the analysis of datasets presented in previous paragraphs. To avoid repetition, we will concentrate on the variability of the basic summary statistics of the results from the nine climate models rather than on their values (these are discussed in previous paragraphs, see also Figs. 3 and 4).

The mid-term mean values of maximum discharge, duration and volume rise with each subsequent time frame (Figs. 5A–5C). Moreover, the rise in mean is steady and consistent and can be observed for both RCP scenarios. Judging by the medians, it can be noticed that the rise is generally significant in the subsequent time frame for all the floodwave properties, that is maximum discharge, duration and volume. For instance, the rise in median of the mid-term mean maximum discharges between baseline and 4.5NF periods is slightly over 6%, between 4.5NF and 4.5FF some go to nearly 16.4%, whereas for RCP8.5 these steps are 12% and 24% high, respectively. The comparative step in medians of the mid-term means between the subsequent time horizons is at least two times higher in duration, and even nine times higher in volume than in maximum discharge. This means that a relatively gradual increase in maximum peak flows (a few percent over a decade) may result in sudden future expansion of the duration and volume of the floodwaves. In general, the quartile values for the mid-term mean are similar to the medians, and their variability around the median does not exceed +/− 50%, except volume, which can even exceed +400% (4.5NF).

Note that it is not only the time-dependent mean that influences the value of a time-dependent parameter, but the time-dependent standard deviation should also be considered (Figs. 5D–5F). However, the situation with the mid-term standard deviation of particular floodwave parameter is more complicated. In general, the median values of the standard deviation are comparable to the values of the mean medians for maximum discharge and duration, whereas they are ca. ten times smaller in the case of floodwave volume. Furthermore, it is impossible to distinguish one single direction for the changes in the mid-term standard deviation with time, because for example the median of the standard deviation of the maximum discharge always rises for the scenario RCP4.5 (ca. by 30% every ~30 years), whereas it drops by 8.5% between the baseline period to the NF time window and rises again by 55% to FF in the RCP8.5 scenario. Such instability in both the magnitude of the difference in subsequent periods and the direction of change can also be observed for duration and volume, but these do not follow the patterns seen in maximum peak flows. This means that the 27-year time frames are perhaps long enough to observe stable gradual changes in the mean of the floodwave parameter values (not always—see the winter season case), but too short to categorically judge their possible variability range, that is standard deviation. The values of the first and third quartiles for mid-term standard deviation are comparable with the equivalent mid-term medians.

As far as the relative trends are concerned (Fig. 6), the box-whisker plots for the mean value (Figs. 6A–6C) clearly indicate that the relative trend in mean for the nine climate models is negligible within subsequent time frames (with the median—thick solid line in a box—closely revolving around 0%) in maximum discharge, duration and volume of the flood waves. The data distribution in datasets (the first and third quartiles) is also generally minimal when compared to the data range, including the outliers visualised by the circles. This means that both the baseline and all four scenarios point to the relative stability of the extreme waters currently as well as in the nearest and further future. The only exception concerns the case of flood volume in the scenario of 8.5FF, where the relative trend in mean reaches nearly 20%, which is probably caused by the projected disproportional increase of the floodwave volume. It is interesting that the resulting graphs are accompanied by distant outliers (especially in duration plots), supporting the idea of great variability of the scenario results created by nine climate models used in calculations.

Figures 6D–6F represents the distribution of the trend in standard deviation in the summer season statistical modelling results. Here, too, the relative trend revolves around 0% for all the cases considered, except Volume 8.5FF in the floodwave volume results, where the relative trend is comparable to the result for the mean value. The distribution of the results is also small – the quartiles reach a maximum of 20%, although a higher range of outliers can be seen, exceeding—500% (for the flood wave duration baseline data). Please note, however, that the short datasets used for estimation and the unambiguous results for mid-term standard deviation suggest that the results concerning trends in standard deviation may be affected by a significant (and unknown) level of uncertainty.

Case B. Winter season—Figs. 7 and 8

The basic summary statistics of the mid-term parameters for the winter season are presented in Fig. 7. The median of the mid-term mean of the duration and volume of the floodwaves increases in each time window for both the 4.5 and 8.5 RCP scenarios (Figs. 7A–7C). The magnitude of this growth varies, however, and ranges from for example 6.5% (duration 4.5NF to 4.5FF) to over 40% (volume 8.5NF to 8.5FF), but usually revolves around 20% (for consecutive time periods). Unlike duration and volume, the mid-term mean tendency for the annual maximum discharges is not monotonic within the entire period considered. The median rises slightly (by 1.3% and 2.7% from baseline to NF, for RCP4.5 and RCP8.5, respectively) and then in FF falls slightly by 5.7% and 7.4% for RCP4.5 and RCP8.5, respectively. Considering the significant range of uncertainty, the mid-term mean annual peak discharges are fairly stable. The upper and lower quartiles for all the floodwave parameters are of a similar value as the median, and they rise by a few percent per decade, which is expected because the uncertainty (and thus the range of the models results) grows with the time distance into the future. To sum up, the results for the mid-term values conform with the analysis depicted in previous chapters (compare Fig. 4).

It is hard to distinguish one monotonicity pattern for the median of the mid-term standard deviation of particular floodwave parameters (Figs. 7D–7F), because it differs not only from parameter to parameter (which can happen), but also between time periods. In addition, the variability of the medians is high, from –137% in volume from baseline to NF, RCP8.5 to +26% in volume from 8.5NF to 8.5FF. The upper and lower quantiles are comparable to the medians, and they increase gradually with the subsequent time series but at a different rate (from 1% to 37%, depending on the floodwave parameter and periods).

As in the case of the summer season, for most of the modelling results and baseline data the relative trend in mean value is small and close to 0% (Figs. 8A–8C). Moreover, it is characterised by generally smaller diversity than can be observed in the summer season. The third and first quartiles do not exceed 2% above or below the median value. In addition, the outliers observed are smaller, apart from volume, where they are nearly—120% high for 8.5NF. All these factors suggest stability of the trend in mean value regardless of the projected scenario.

The pattern revealed in the graphs for the mean is repeated in the plots for standard deviation (Figs. 8D–8F). The trends are relatively small—close to 0% and their distribution is practically negligible for all scenarios, suggesting stability of the amplitude of extreme flows in the analysed cross-section.

Figure 9 shows the relationship between maximum discharge (Qmax) at Osowiec gauge and maximum flood area, estimated on the basis of the results for the period 1961–1996 described in Chormański, Mirosław-Świątek & Michałowski (2009). Average time (days) of inundation in LBB vegetation communities (Fig. 1) vs. duration of flood at Osowiec gauge are shown in Figs. 10–13. This analysis (Fig. 9) revealed a functional relationship between maximum inundation area and Qmax. The two outliers are related to the maximum inundation in this area that occurred in 1979 and the minimum value of Qmax in the time series that was recorded in 1984. The relationship between the duration of peak flows and the average duration of inundation in the abodes of CA, CG, GM, PC indicates a strong linear relationship with the correlation coefficients R = 0.977, 0.906, 0.827, 0.865, respectively.

Figure 9 Maximum discharge (Qmax) at Osowiec gauge vs. maximum flood area.

NR, number of raster in DTM.

Figure 10 Average time (days) of inundation in Caricetum appropinquatae vs. duration of flood at Osowiec gauge.

Figure 11 Average time (days) of inundation in Caricetum gracilis vs. duration of flood at Osowiec gauge.

Figure 12 Average time (days) of inundation in Phragmitetum communis vs. duration of flood at Osowiec gauge.

Figure 13 Average time of inundation in Glycerietum maximae vs. duration of flood at Osowiec gauge.

The lower values of the correlation coefficient R (time of inundation vs. duration of flood) for GM and PM communities probably may result form a larger spatial heterogeneity in elevation within these communities (Fig. 1) but also from the uncertainty of the modelled flood-water table in the valley stemming from use of the 1D unsteady flow model (by Chormański, Mirosław-Świątek & Michałowski (2009)) and the insufficient resolution of the DEM. However, on a landscape scale it has been shown that river floods, water quality and vegetation are strongly interlinked in the Biebrza floodplains (Keizer et al., 2016).

Discussion

In the current study, we estimated the mean changes of flood indices (30-year mean values) in the near and far future (NF and FF) with respect to the reference period (Baseline). It is widely recommended to use a 30 year period in describing the impact of climate change, as it takes approximately 30 years for any difference between current emission scenarios to have an noticeable impact on the climate (Carter, 2007). Such an approach is commonly used in climate change impact studies instead of long-term trend analyses which are characterized by non-stationarity (Meresa, Romanowicz & Napiorkowski, 2017; Razavi et al., 2016). Furthermore, for a proper consideration of climate impacts the sub-periods analysed should not contain clear trends. In this study an annual trend analysis of selected flood indices was therefore performed for each of the projection periods (Baseline, NF and FF). The statistical analyses (Figs. 5 and 6) indicate that no significant trends occur in the time series of the duration and volume of floods or in their culmination in the historical period and future projections, which could be caused by a gradual increase or decrease in the value of these indicators as a result of climate pressure. For comparison, the results obtained by Maksymiuk et al. (2008) for the Burzyn water gauge (closing the River Biebrza’s Lower Basin) indicate that the trend in maximum annual discharges was negligibly decreasing in the period 1966–2003.

The results presented in this study clearly indicate that both volume and duration of winter floods will increase continuously with the time horizon and from RCP4.5 to RCP8.5. At the same time, the reduction in peak annual floods is expected to decline slightly in all scenarios. Duration and volume of summer floods seem to follow the same pattern as for the winter season, however, the projected magnitude of increase is much lower. Such a discrepancy in hydrological response is most probably caused by the expected precipitation and temperature changes. In the future time horizon, precipitation is expected to increase more intensively in winter compared to summer. Additionally, temperatures in winter season are set to increase causing a decrease in snowfall and consequently a reduction in the volume of water originating from snow thawing during the spring. This explains the reduction of peak flows usually occurring during the spring thaw in the Biebrza catchment. A slight increase in precipitation and temperature during the summer season causes an increase in evapotranspiration and consequently the overall increase in summer floods is moderate.

In this study, flood characteristics (duration, volume and maximum discharge) were analyzed in terms of different emission scenarios and time horizons. A similar investigation has been conducted on a different scale (Europe) by Schneider et al. (2011). In their study, the authors assessed flooding by quantifying the changes in magnitude, timing and duration of overbank flows for major European rivers affected by climate change, using the global scale hydrological model WaterGap (Water–Global Assessment and Prognosis) run by three climate change projections (IPCC SRES—Special Report on Emissions Scenarios—A2 and the SRES B1) for the year 2050. Their results show that for Central and Eastern Europe, flood volume causing floodplain inundation is expected to be reduced in the 2050s. A similar outline is drawn for flood duration which also seems set to decline in the future. In another study by Schneider et al. (2013), a decrease in flood volume was predicted for inundation and duration of overbank flows in Central and Eastern European rivers. In addition, projections carried out by Dankers & Feyen (2008) found a significant decrease in the occurrence of flood hazards in the northeast of Europe. It is noteworthy that the current study presents opposite results that is volume and duration of floods are projected to increase continuously in the future time horizons. It must be noted that there are numerous reasons why hydrological projections differ, so a direct comparison to other studies is usually hampered with methodological differences (Kundzewicz et al., 2017)—first of all, the scale of the hydrological model, which in the case of Schneider et al. (2011, 2013) and Dankers & Feyen (2008) covered the whole European continent, compared to the catchment scale Vistula and Odra Model. The model scale directly entails the resolution of climate change projected data and hence, results for the Biebrza Valley presented in the scale of the whole European continent is far more uncertain In addition, the climate models themselves vary significantly according to the GCM/RCM used and the downscaling method applied. Dankers & Feyen (2008) used only one RCM, whereas Piniewski et al. (2017a) used 9 RCMs and draw conclusion based on their median response.

Our results revealed that for the three considered time periods, that is 1971–2000 (historical period) and future projections 2021–2050 (NF), 2071–2100 (FF), the trends in mean and standard deviation related to the respective base values (see Figs. 6 and 8) in the time series of the duration and volume of floods or in their culmination are close to zero percent. It means that these hydrological parameters of floodwaves are expected to remain fairly stable and no noticeable increase or decrease in these parameters is anticipated. In consequence, this implies that the wetland is not under threat from climate change in the sense that the wetland will disappear or transit to a fundamental other state (dryer, wetter) which can be regarded as good news for the stability of the floodplain ecosystem as a whole in the Lower River Biebrza Basin. This is an important finding suggesting that sustainable preservation of the wetland is likely from a hydrological perspective and that the function as an important stopover site for migratory birds related to wetlands will continue (Budka et al., 2019; Maciorowski et al., 2014; Polakowski, Kasprzykowski & Goławski, 2018).

However, the risk of malfunctioning of riparian wetlands is associated with an almost twofold increase in the duration of summer inundations, which increases with the time horizon (a greater increase in the distant future than in the near one) (Fig. 4). The duration and volume of winter floods seem to follow the same pattern as for the summer season (Fig. 3). Figures 10–13 indicate that the durations of inundation in plant communities in the Lower River Biebrza Basin are strongly correlated with the duration of flood monitored at the Osowiec gauge. This implies that all of these plant communities will face a significant change in duration and inundation depth both during winter and summer inundations. Apart from changes in habitat conditions (Kotowski et al., 2006) this will hamper the possibility for mowing and harvesting of hay from wetlands. Timely mowing is particularly important to preserve low nutrient availability and low growing herbaceous vegetation (Olde Venterink et al., 2009). Failure to mow in large areas of open peat bogs may result in a succession of shrubs and forest communities (Piórkowski, 2003), which would consequently transform the open marshes of floodplains towards ecosystems with lower biodiversity. The disappearance of valley meadows that could be maintained by continuing periodic mowing (Mioduszewski & Ślesicka, 2004) would also result in the loss of rare wetland birds such as Aquatic Warbler, Ruff, Great Snipe and Greater Spotted Eagle. However, it should be noted that although seasonal mowing is a positive management factor for the biodiversity of non-forested wetlands, analysis of the effects of rotational wetland mowing on breeding birds provided evidence for a positive effect on some wetland birds of significantly longer mowing intervals (Antoniazza et al., 2018).

Grygoruk et al. (2014) assume for the time horizon 2070–2100 a significant increase in precipitation in summer, a decrease in precipitation in autumn, winter and spring and a general increase in the average air temperature and on this basis, they present hypothetical scenarios of changes in flood condition and their ecological consequences in the Lower River Biebrza Basin. These assumptions were made on the basis of the GCM–RCM (combined with the SRES A1B emission scenario) projections on temperature and precipitation changes. Contrary to this hypothetical approach, in our research we used flow hydrographs simulated with the calibrated and validated SWAT model and considering nine bias-corrected climate models with the modern generation emission projections (RCP) 4.5 and 8.5.

The increase in maximum discharges observed for summer floods in both the near and distant future (Fig. 4) leads to an increase in flood extent (Fig. 9). This may lead to extension of the zone of flood tolerant communities, such as Phragmitetum, GM and CG on cost of less flood tolerant communities such as CA and Calamagrostietum strictae. If the flood zones would extend in prolonged periods to the valley edge moss-sedge communities such as Scorpidio Caricetum elatae and Caricetum rostrato-diandrae might disappear (Palczynski, 1984; indicated in Fig. 1 as mixed sedges and mixed sedges and grasses). In such a case these ecologically valuable low sedge meadows might be displaced by taller growing sedges that have a larger tolerance to prolonged flooded conditions (Wassen, Peeters & Olde Venterink, 2002), which in consequence would lead to loss of mosses, forbs and low sedges and transformation towards more productive ecosystems of lower biodiversity (Palczynski & Stepa, 1991; Swiatek et al., 2008).

The extension of the duration of inundations while temperatures are increasing as spring advances is expected to lead to higher nutrient availability, since the turnover of organic matter is expected to increase in warmer standing water (Wassen et al., 2006). Currently already algal blooms occur during spring inundations when spring temperatures are relatively high (personal observation Wassen). This may occur more frequently in future leading to decreased transparency of the flood water which in turn may hamper the growth of mosses and forbs that normally start growing while inundations still occur. The increase in nutrients may also be dangerous for fish, because when coinciding with a surge in higher flow durations, the excess of nutrients can inhibit fish reproduction. The nearly double duration of winter inundations (a greater increase was observed in the distant than in near future) (Fig. 3) may also hinder the removal of bushes (which is normally done in winter season) and cause difficulty in mechanically mowing in large swamp areas, which in turn is expected to lead to loss of typical herbaceous floodplain vegetation.

On the other hand, the extension of the duration and volume of floods (which occurs both in winter and summer inundations) is a positive phenomenon because it stretches the period of watersaturated and moist conditions of hydrogenic habitats and prevents over-drying. Dropdowns in water tables in the floodplain in the growing season were shown to be highly correlated to a boost in nitrogen release (Wassen, Peeters & Olde Venterink, 2002; Wassen & Olde Venterink, 2006). Thus, if water saturated conditions prevail longer in summer, we expect nitrogen availability to decrease. Furthermore, prolonged inundation will slow down decomposition of plant remains and prevent degradation of the peat substrate, which is of great importance for maintaining the functioning of the wetlands especially the fens on largely organic substrate (Mirosław-Świątek et al., 2017). Moreover, the accumulation of peat may increase when water saturation extends over longer periods.

The results of statistical analyses quantifying the impact of climate pressure on selected hydrological indices of floods (duration, maximum flows, volume of floods) imply that the effectiveness of the swamp, marsh and fen ecosystem management may be limited due to a failure to adapt these activities to climate change, and that new methods may need to be introduced to protect nature in the area of the lower River Biebrza basin (Grygoruk & Rannow, 2017).

Conclusions

The trend analysis of mean and standard deviation of peak flows, floodwave duration and volume for both observed and modelled scenarios revealed negligible tendencies in the scrutinised datasets for the summer and winter hydrological seasons within the three time windows analysed. This implies that the wetland is relatively stable under climate change.

The results presented in this study clearly indicate that both volume and duration of winter floods will increase continuously with the time horizon and intensification of RCP. At the same time, the reduction in peak annual floods is expected to decline slightly in all scenarios. Duration and volume of summer floods seem to follow the same pattern as for the winter season, however, the projected magnitude of increase is much lower.

Despite the fact that statistical analyses showed no trend of the hydrological characteristics analyzed (maximum discharge, duration and volume) within the analyzed time windows, the increase in flood duration as a consequence of climatic pressure in the near (2020–2050) and distant (2070–2100) future points to the possibility of difficulties in maintaining the natural diversity and agricultural use of wetlands in the LBB. Changes in environmental conditions (e.g. an almost two-fold increase in the duration of floods in the near and distant future) and other environmental changes may lead to considerable shifts in transversal ecosystem zonation with expansion of flood-tolerant communities such as Phragmitetum and CG on cost of less flood-tolerant communities such as CA and other low-growing sedge-moss communities. Furthermore, nutrient availability and algal blooms may increase during spring inundations but lower turnover of organic matter later in summer may lead to a higher peat accumulation rate. Logistical problems with summer mowing and removal of bushes in winter may prevent the preservation of selected habitats which are under the management of the Biebrza National Park.

Supplemental Information

Supplemental Information 1 Daily discharge at Osowiec gauge.

Click here for additional data file.

Supplemental Information 2 SWAT model simulated discharge time series used for trend analysis.

Click here for additional data file.

Additional Information and Declarations

Competing Interests

Author Contributions

Data Availability

The authors declare that they have no competing interests.

Dorota Mirosław-Świątek conceived and designed the experiments, performed the experiments, analyzed the data, prepared figures and/or tables, authored or reviewed drafts of the paper, and approved the final draft.

Paweł Marcinkowski conceived and designed the experiments, performed the experiments, analyzed the data, prepared figures and/or tables, and approved the final draft.

Krzysztof Kochanek conceived and designed the experiments, performed the experiments, analyzed the data, prepared figures and/or tables, and approved the final draft.

Martin J. Wassen conceived and designed the experiments, analyzed the data, authored or reviewed drafts of the paper, and approved the final draft.

The following information was supplied regarding data availability:

Model simulated discharge data of Osowiec gauging station for different climate change projections (RCP4.5, 8.5) are available in the Supplemental File.

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
