# Peer review of "The impact of climate change on flow conditions and wetland ecosystems in the Lower Biebrza River (Poland)"

_PeerJ, doi:10.7717/peerj.9778_

## Round 0.1 · original submission · Major Revisions

This manuscript has been reviewed by two experts in this research field. Both reviewers suggest a major revision. Please respond to their comments in detail and I will ask them to re-review it.

Reviewer 1 ·

Basic reporting

I think the literature review/methodology/results about the SWAT model is missing or very poor.

Experimental design

There is no description about the Calibration and Validation of the SWAT model.
The authors should be explained which period have used for the calibration/validation of the SWAT model?
How did they calibrate the SWAT model? Which environment/software has been used for calibrating?
There is not any literature review about the SWAT model. For example, the following paper has been recently published:
Choubin et al., 2019. Streamflow regionalization using a similarity approach in ungauged basins: Application of the geo-environmental signatures in the Karkheh River Basin, Iran. Catena

Validity of the findings

Validation results of the SWAT model are missing.
Uncertainty analysis is missing, too. R-factor and P-factor are two statistics that are used to investigate uncertainty in the SWAT model when calibrating it by the SUFI2 algorithm (Choubin et al., 2019).

Reviewer 2 ·

Basic reporting

Clear and unambiguous, professional English used throughout.
The paper is well-written and professional English language is used throughout

Literature references, sufficient field background/context provided:
Literature references are appropriate for the statements made in the introduction. However, the background provided in the introduction does not fit to the analyses made. Two examples: Lines 18-22 highlight the importance of feedback relationships between human activities, climate change and ecological dependencies - however these feedbacks have not been made clear in the analyses. In lines 30-31 it is mentioned that spring melt is the main inundation driver. In the paper, only lines 466-470 in the conclusions refer to this important dependency. Especially since the results show opposite tendencies as comparable studies, this should have been investigated in detail.

Professional article structure, figures, tables. Raw data shared:
The structure of the article the authors were aiming for is clear, but the content is not appropriate. One example: The investigations of inundation are presented in the discussion section without having been explained in the methods or shown in the results.
Figures are well designed, Figures 4 to 7 require larger font to be clearly readable. No table is provided.
Raw data shared is accessible, but lacks explanation. It remains unclear from what analysis step the data is provided. Is this the raw SWAT model output at the gauge Osowiec? What does the entry "IMGW" mean? Where is the data you mention is being used for the analysis (line 59) ranging from 1955-2011? Mention somewhere that only flows higher 25m³/s are provided. It would have helped to also provide the three datasets mentioned in lines 125-129.

Self-contained with relevant results to hypotheses:
I have difficulties to relate the results to the aims given in the introduction (no hypotheses are given). The results are not well structured according to these aims. I would have expected to see the methods and also results structured according to: 1. trends in duration, volume and max discharge; 2. change in flood characteristics with respect to time horizon (what is actually meant by that? it is not explained in the paper); 3. same as 2 but also for seasonality; 4. Inundation and potential changes to the ecosystem.

Experimental design

Original primary research within Aims and Scope of the journal:
It is written in the "Aims&Scope" of PeerJ: "PeerJ only considers Research Articles and Literature Review Articles. It does not accept Hypothesis Papers, Commentaries, Opinion Pieces, Case Studies, Case Reports.". But the title of this article (and the content, too) mark the work as a "case study" and hence does not seem eligible for PeerJ.

Research question well defined, relevant & meaningful. It is stated how research fills an identified knowledge gap:
There is a lack of well defined research questions and knowledge gaps. One could argue, that (1) anthropopressure, line 12-15; (2) feedback relationships, line 20-22; (3) relationship between spring thaws and melt from precipitation, lines 30-31 and (4) climatic pressures impact on ecosystem management, lines 38-39 are meant to be knowledge gaps, but those are not tackled in the manuscript.
Further, it is stated in the introduction that "In this article, the impact of climate change on the hydrological characteristics of floods (duration, volume, maximum flow) was quantified". The impact of climate change on hydrological characteristics is not a knowledge gap - many studies are available carrying out such research and the present work provides a case study of this topic.

Rigorous investigation performed to a high technical & ethical standard.
Partly true. As mentioned above, the gaps and aims defined are not dealt with properly.

Methods described with sufficient detail & information to replicate.
No. I am unable to replicate the work. Especially the whole trend analysis section lacks detail and information to be replicable and I miss the explanation of the purpose. It is impossible for me to replicate or understand how figures 4-7 were produced. I do not understand why you need these statistical distributions when all the data you need for the analysis is already contained in the time series of the SWAT models. Please explain that in detail.
The inundation analysis is not described in methods (and also not in results) - How Figs 8-12 are produced is not explained in methods.

Validity of the findings

Impact and novelty not assessed:
Yes

Negative/inconclusive results accepted:
Not applicable

Meaningful replication encouraged where rationale & benefit to literature is clearly stated
Not applicable

All underlying data have been provided; they are robust, statistically sound, & controlled:
Yes

Conclusions are well stated, linked to original research question & limited to supporting results, Speculation is welcome, but should be identified as such:
It is difficult for me to understand the results since the methods lack detail and information to reproduce Figures 4-7.

I understand why the authors conclude that the wetland is impacted negatively and may be under threat. However, based on what I learned from the wetland description in the paper, I do not agree with the solely negative ecological impact of increasing inundation duration and extent. If inundation increases in a wetland, yes, conversion of certain vegetation types are occurring (e.g. as stated from sedge meadows to reeds or bush and wood to meadows or wetland grasses) but this can eventually also lead to more habitat diversity and the wetland covering a larger area (e.g. fringe vegetation of meadows is shifting further outwards). Testing these conclusions requires e.g. a distributed hydraulic model to check which areas get inundated with which depth and duration which may lead to a possible shift of wetland vegetation areas over longer time periods. Or you need to explain why this shift/any positive effect is unlikely / not possible to happen.

Additional comments

Title and Abstract
"impact of climate change on wetland ecosystems" is not reflected in the abstract, where you mention changes in flow conditions only. The impact on the wetland needs to be mentioned in the abstract, otherwise the title is misleading. Also, the impact on wetland ecosystem is only described in the discussion and has not been investigated in detail across the papers methods and results.

Introduction:
- You cite studies directly related to your work, e.g. lines 38-48 and line 52. I suggest to link your discussion to e.g. Grygoruk and Rannow (2017) and how one can deal with forecasted climate changes and maybe even translate it into positive. Also, what did the tests of Maksymiuk reveal compared to your historical analysis?
- I suggest to clearly define research gaps throughout the introduction, formulate research questions and aims at the end of the introduction (see comments above) and use this structure also for methods and results. Currently, the reader is lost which method and results relate to which aim.

Study area:
Line 71: Add the size of the wetland.
Also, add the main flooding mechanisms (is it solely governed by inflows or is there also an impact of precipitation or groundwater, for instance).

SWAT model:
While the SWAT model is generally well calibrated for the Vistula and Odra river basins, it is required to give information for the applicability for this study: What is the performance (calibration and validation including time periods used), e.g. use the KGE and its three components that were used as performance criteria in Piniewski's paper for the Osowiec gauge. Also, how does the model perform in depicting the snowmelt mechanisms (e.g. in dry, wet, warm and cold years) which are of extreme importance when analysing climate change impacts in your study area?

Climate Change Scenarios:
It would help to provide a figure that shows the actual time series (maybe even in supplementary material) of the three datasets for which trend tests are carried out.

Trend Analysis - the method
I do not understand large parts of this section. Why don't you just extract the annual SWAT results and then run an Anova test on the three datasets to see if the change is significant or not?
l.138-145: water management in the modelled catchment: Any management that impacts peak, volume and duration is considered in your analysis, or?
l.146-150: Why gaps? You use modelled data. There should be no gaps?
l.157-158: Why are these distributions used because they can "model annual and seasonal peak flows"? I understand that SWAT was used to model annual and seasonal peak flows. This leads to further confusion throughout the paper: I don't understand the term 'models' in your paper. Once you seem to refer to the 9 climate "models" used to drive SWAT, then you refer to the statistical "models" developed. This is extremely confusing.
l.177-205: I do not understand this section at all and why it is needed in your analysis. Number equations in l.183-184. Why do you have 5 non-linear equations? What are the '?' in Eq.1? Can you provide an example of what you are actually doing in the supplementary material? (e.g. show the SWAT data and then the fitted distributions)
l.200-201: 'we developed software' but this is referenced to another author? What software was developed and where can it be accessed? The software can now cope with measurement gaps? Why is this needed? You analyse only modelled data (at least I could not find observed data in the raw data provided)?

Results
Fig 2 and 3: Explain what the rectangle and the box mean in the boxplot (probably mean and median)
l.228-229. I do not understand this sentence at all.
l.229 ff. It needs to be made clearer in the methods. I am unable to understand this.
l.248ff. Why did you choose the 'mid-term year' (14) to analyse the results? You neglect the natural variability here by picking one arbitrary year from the 30 year time series? Why not considering all years? Why is this not properly described in methods? I cannot reproduce how Figures 4-7 are produced.
l.281ff. My lack of understanding the methods makes it impossible to understand these results. Also, I suggest to structure the results according to the aims in the introduction. In the raw data winter months are given as numbers from 1-6, summer discharges from 7-12. Does it mean the year is divided into January - June = Winter and July - December = Summer? Clearly, this is not an appropriate distinction of north-eastern European seasonality.

Discussion
l.371-388: This is a mix of methods, results and discussion. As mentioned earlier, I suggest to structure the article properly according to the research questions to be defined in the introduction.
line 397-400. Fig 4 is not a trend figure. Also, I see a trend for volume of floods for 85FF in Fig 5. But what do you mean with "significant"? Were any significance tests carried out?
Fig 5 what do "dominating models" mean? Again, confusion between the 9 climate models and the statistical distribution functions (also termed "models") (in brackets).
l.407ff: "seasonal mowing" (l.79) is highlighted as a positive factor for the wetland. However, please note that significantly longer mowing intervals are better (Antoniazaa et al. 2018)
l. 416-418: I don't understand. Figure 3 is not about an increase of discharges in near and distant past and Figure 6 not about flood extent.
l.423: Why is there an increase in nutrient content?
l.451-452: I understand the reasons you give that lead to different results in different studies. But is it really not possible to investigate reasons for having totally opposite results, e.g. is it due to different climate change data (increase vs decrease in pcp)? If this is similar, do the hydrological models give non-linear results so that SWAT is different to the models in the other studies?


References:

Antoniazza, M., Clerc, C., Le Nédic, C. et al. Long-term effects of rotational wetland mowing on breeding birds: evidence from a 30-year experiment. Biodivers Conserv 27, 749–763 (2018). https://doi.org/10.1007/s10531-017-1462-1

---

## Round 0.2 · Minor Revisions

Please modify your manuscript based on review comments, especially Reviewer #2, who provides detailed comments on how to improve the quality of your manuscript.

Reviewer 1 ·

Basic reporting

It has been revised

Experimental design

It has been revised

Validity of the findings

It has been revised

Reviewer 2 ·

Basic reporting

no comment

Experimental design

no comment

Validity of the findings

no comment

Additional comments

The authors have revised the manuscript in detail! Thank you for considering the comments and for the extensive explanations.

I observed the following points that the authors may want to consider before publication:

l.30-31 vegetation shifts and nutrient availibility and algal blooms: This is not particularly backed up by modelling or observation, right? And this is based on your discussion and in my view should also be treated in the abstract as such. So, be more cautious by adding "will likely take place" "This may lead to..."
l.108 either Lower Biebrza River or Lower Biebrza Basin
l.109 suggest not to refer to figures in the introduction, just write: "...related to the characteristics of floods."
l.182 by snowmelt processes
l.184 Fig 2A: The KGE of 0.8 for the validation period seems strange. The fit is visually worse than the baseline, the r2 is lower, but KGE is higher, please double check the calculation
l.184 one of the wettest
l.300-304 this sentence requires language correction and suggest to split it in two sentences
l.335 ...revealed that a certain dataset, i.e. ....
l.341 bestaccording (space missing)
l.408 please explain LBB on first use
l.487 strong linear relationship with the correlation coefficients R=...
l.493 you use DEM and DTM throughout your text. In case you used only one, please stick to the correct one.
l.514 "intensification of RCP" - better to rephrase to "and from RCP4.5 to RCP8.5"
l.525 moderate
l.566 Osowiec gauge. (and subsequent occurences of "water gauge")
l.589-590 this does not necessarily lead to lower uncertainty. I suggest to remove that sentence. You have mentioned that you are using a different forcing dataset, so that is fine.
l.651 gracilis, possible replacing less flood.... ?

comments:
It would help to provide a figure that shows the actual time series (maybe even in supplementary material) of the three datasets for which trend tests are carried out.
It has been added to the supplementary material.
-> I cannot see this figure in the supplementary material, the supplementary material is made of one Excel table.

l.157-158: Why are these distributions used because they can "model annual and seasonal peak flows"? I understand that SWAT was used to model annual and seasonal peak flows. This leads to further confusion throughout the paper: I don't understand the term 'models' in your paper. Once you seem to refer to the 9 climate "models" used to drive SWAT, then you refer to the statistical "models" developed. This is extremely confusing.
In the paper we use SWAT for crating the datasets for further analysis of trends. The statistical models (or probability distribution functions) were used for parametric calculation (modelling) of the trends in mean and standard deviation in time series ‘produced’ by SWAT. The term ‘statistical model’ is the approved synonym of probability distribution function in hydrological sciences, especially that the latter is mathematical rather than the hydrological term.
-> Yes, I understand these differences. But you need to make sure that every one of these three aspects are named differently. These three components of your work cannot be named just "model" because the reader then does not know about which you are referring to. Stick to the proper names (climate model, SWAT or hydrological model, statistical model) even though it means a few more words


l.451-452: I understand the reasons you give that lead to different results in different studies. But is it really not possible to investigate reasons for having totally opposite results, e.g. is it due to different climate change data (increase vs decrease in pcp)? If this is similar, do the hydrological models give non-linear results so that SWAT is different to the models in the other studies?
The differenced in results between these studies comes from two main facts:
1) Difference in model resolution – LISFLOOD model (Dankers and Feyen, 2008) was set-up for the whole European continent and VOB (Piniewski et al., 2017) for two Polish basins (Vistula and Odra). Results for the Biebrza Valley presented in the scale of the whole European continent is far more uncertain.
2) Difference in climate projections representation - Dankers and Feyen (2008) used only one regional climate model, which is not in line with the IPCC recommendations (2014) to use the ensemble of models in order to decrease the uncertainty of predictions. Piniewski et al. (2017) used 9 Regional Climate Models and draw conclusion based on their median response.
-> I suggest to briefly (e.g. max one sentence each) mention these reasons in the text so that the reader know why these differences occur.

---

## Round 0.3 · accepted · Accept

I am happy to accept your manuscript, and thanks again for submitting it to PeerJ! However, please pay attention to typos as Reviewer 2 mentioned during proofreading stage.

Reviewer 2 ·

Basic reporting

no comment

Experimental design

no comment

Validity of the findings

no comment

Additional comments

Thank you for the second round of revisions.

I only noticed a typo in your corrections (there may be more in the paper, please carefully check the print proof):

l.300-304: ...were descripted in details in the Chormański article (Chormanski et al., 2009)...
correct to ...were described in detailn in Chormanski et al. (2009)...

It is not needed to return the paper to me and it is acceptable now in my opinion. Congratulations for your paper.